# Comparative Plastomes of *Curcuma alismatifolia* (Zingiberaceae) Reveal Diversified Patterns among 56 Different Cut-Flower Cultivars

**DOI:** 10.3390/genes14091743

**Published:** 2023-08-31

**Authors:** Jie Wang, Xuezhu Liao, Yongyao Li, Yuanjun Ye, Guoming Xing, Shenglong Kan, Liyun Nie, Sen Li, Luke R. Tembrock, Zhiqiang Wu

**Affiliations:** 1College of Horticulture, Shanxi Agricultural University, Jinzhong 030801, China; wj18434764189@163.com (J.W.); xingguoming@163.com (G.X.); saulisen@163.com (S.L.); 2Shenzhen Branch, Guangdong Laboratory for Lingnan Modern Agriculture, Genome Analysis Laboratory of the Ministry of Agriculture, Agricultural Genomics Institute at Shenzhen, Chinese Academy of Agricultural Sciences, Shenzhen 518120, China; liaoxuezhu@caas.cn (X.L.); liyongyao@caas.cn (Y.L.); kanshenglong@caas.cn (S.K.); nieliyun18@163.com (L.N.); 3Guangdong Provincial Key Lab of Ornamental Plant Germplasm Innovation and Utilization, Environmental Horticulture Research Institute, Guangdong Academy of Agricultural Sciences, Guangzhou 510640, China; yeyuanjun199910@126.com; 4Department of Agricultural Biology, Colorado State University, Fort Collins, CO 80523, USA

**Keywords:** plastomes, comparative analyses, phylogeny, ornamental horticulture, *Curcuma alismatifolia*, siam tulip

## Abstract

*Curcuma alismatifolia* (Zingiberaceae) is an ornamental species with high economic value due to its recent rise in popularity among floriculturists. Cultivars within this species have mixed genetic backgrounds from multiple hybridization events and can be difficult to distinguish via morphological and histological methods alone. Given the need to improve identification resources, we carried out the first systematic study using plastomic data wherein genomic evolution and phylogenetic relationships from 56 accessions of *C. alismatifolia* were analyzed. The newly assembled plastomes were highly conserved and ranged from 162,139 bp to 164,111 bp, including 79 genes that code for proteins, 30 tRNA genes, and 4 rRNA genes. The A/T motif was the most common of SSRs in the assembled genomes. The *Ka*/*Ks* values of most genes were less than 1, and only two genes had *Ka*/*Ks* values above 1, which were *rps15* (1.15), and *ndhl* (1.13) with *petA* equal to 1. The sequence divergence between different varieties of *C. alismatifolia* was large, and the percentage of variation in coding regions was lower than that in the non-coding regions. Such data will improve cultivar identification, marker assisted breeding, and preservation of germplasm resources.

## 1. Introduction

*C. alismatifolia* is a species in the Zingiberaceae family and native to Southeast Asia [1]. Also known as the “summer tulip”, this species was introduced into China in the 1990s [2]. *C. alismatifolia* has been well known in Asia through its long history as a medicinal plant [3,4]. The showy part of most Zingiberaceae species are bracts, and because of the numerous and aesthetically attractive bracts generated by various cultivars, species in *Curcuma* have recently gained popularity among floriculturists [5]. The ornamental cultivars of *Curcuma* are often of hybrid origin, with polyploidy common among many wild and cultivated lineages [6,7]. The traditional diagnostic characteristics used to identify *Curcuma* are insufficient and difficult to utilize because floral morphology, rhizome color, bract shape and color, and position of inflorescences in *Curcuma* are neither universal or unique to all species [8]. In recent years, increased interest in *Curcuma* has motivated researchers to employ molecular techniques to better resolve relationships between different cultivars and species [9]. The use of molecular techniques is further justified by the fact that morphological characteristics used to identify *Curcuma* lineages often do not resolve closely related lineages. Additionally, the lack of high-quality nuclear reference genomes and the presence of hybrid lineages in *Curcuma* make the use of plastomic techniques especially applicable for the resolution of relationships in the immediate term. The species *C. alismatifolia*, which is one of the largest and most showy in the genus *Curcuma*, contains many named cultivars with diverse hybrid origins. Such complex pedigrees can make the tracing of phylogenetic relationships difficult and potentially hinder breeding efforts. While some plastome data have been generated for *Curcuma* [6,10], no previous work has used this type of data to study the relationships between *C. alismatifolia* accessions. Such efforts will enrich our understanding of the history and diversity of *C. alismatifolia* cultivation as well as improve future breeding efforts through avoidance of incompatible crosses.

Through the swift advancement of sequencing technology and related cost reductions, the plastid genome or plastome, which is simpler and cheaper to assemble than the nuclear genome, is now regularly employed in phylogenomic and population genomic studies [11,12,13]. In nearly every case, plastids in land plants, which are typically between 100 and 200 kb in length, have a typically circular structure, including two inverted repeats (IRa/b), a large single-copy (LSC) region, and a small single-copy (SSC) region between them [14,15,16]. Plastomes are inherited through the maternal line and lack recombination in nearly all angiosperm lineages [17,18]. The small genome size, lack of recombination, and conserved genomic structure of the plastome are the main factors contributing to the simplicity and accuracy of plastome genome assembly and analysis [19,20]. For precise species identification and phylogenetic inference, plastome genome sequences have been extensively exploited [21,22]. Comparative plastome genomics can also uncover mutational hotspots, which provide robust signals in species differentiation, phylogenetic analysis, and population genetic studies, as well as crucial information on the study of plastome evolution, such as patterns of gene loss and IR border variability [23,24]. While mutational hotspots are widely used in barcoding projects, entire plastomes are now being employed as super DNA barcodes given the relative ease in generating such data [25,26,27]. Comprehensive characterizations of numerous individual plastomes within a species form the basis of functional genomic investigations and can guide efforts in chloroplast transformation in efforts to improve metabolic function [28]. Additionally, with further genomic classification and comparison, the plastome can be used as a substantial repository of molecular markers for mapping, phylogenetic study, population-level research, and genetic analysis [29,30,31]. Lastly, the resolution of plastome lineages is necessary for plant breeders in characterizing plastome–nuclear–genome incompatibilities such that inviable crosses can be avoided in hybrid cultivar development [32,33].

Because of the reticulate history and widespread morphological diversity found in *C. alismatifolia*, this study aimed to (1) contribute full-length plastomes from previously uncharacterized lineages of *C. alismatifolia* cultivars, (2) perform comparative analyses from these plastomes to more effectively comprehend the evolution of genome architecture at the intraspecific level, (3) reconstruct the maternal phylogenetic relationships in *C. alismatifolia* using plastome evidence and compare this to previous taxonomic delimitations including named cultivars, and (4) propose novel DNA markers to discriminate *C. alismatifolia* maternal lineages. We expected to find plastomic introgression from species outside of *C. alismatifolia* if interspecific crosses were involved in the history of the cultivated lineages. Furthermore, we expected to find high levels of genomic diversity within *C. alismatifolia* given the high levels of morphological diversity between different cultivars.

## 2. Materials and Methods

### 2.1. Sampling and DNA Extraction, Sequencing

Fresh plant leaves from about 56 accessions of *C. alismatifolia* were collected for DNA extraction (Appendix A). We placed the collected fresh plant leaves of the samples in −80 °C. The material for these extractions was collected from plants growing at Guangdong Academy of Agricultural Sciences. An established cetyl trimethyl ammonium bromide (CTAB) method was utilized to extract total genomic DNA before Illumina paired-end sequencing [34].

### 2.2. Plastome Sequencing, Assembly, and Annotation

We first extracted DNA from libraries with an insert length of 500 bp and sequenced the paired-end read length of 150 bp using MGISEQ-T7. We used Fastp v0.20.1 [35] for quality control filtering of the raw data, applying the following criteria, which produced around 6 Gb of clean reads for each accession: filtered reads with adapters, reads with N bases greater than 10%, and reads with low-quality bases greater than 50%. The chloroplast database was used to match all paired-end clean reads via the bwa v0.7.17-r1188 [36] software. Plastome-specific reads were then chosen using Picard v2.20.3 [37]. The selections of plastome reads were assembled using SPAdes v3.14.0 [38] with default parameter settings, and the resultant scaffolds (GFA files) were imported using Bandage v0.8.1 [38] to generate the complete plastome for each accession. The *C. phaeocaulis* plastome was used as a template to give a complete reference. All genes were manually delimited. The Plastid Genome Annotator (PGA) [39] was then used to perform a complete re-annotation of every species, followed by manual revisions where necessary. The Draw Organelle Genome Maps online software (OGDRAW version 1.3.1) [40] was used to implement the depiction of plastome structure.

### 2.3. Comparative Genomic Analysis

REPuter typical settings were used to identify four distinct repetition types, including forward, reverse, palindrome, and completion [22,41]. The Perl script MISA was used to find simple sequences repetition (SSRs), with 10, 6, 5, 5, 5, and 5 repeat units set for mono-, di-, tri-, tetra-, penta-, and hexa-motif microsatellites as the minimum threshold, respectively [42]. Using the Ca1 plastome as a reference, genome sequence diversity analysis was conducted using mVISTA to compare 11 representative plastomes using the Shuffle-LAGAN [43] alignment tool. To determine the nonsynonymous (*Ka*), synonymous (*Ks*), and the ratio of nonsynonymous to synonymous nucleotide substitutions (*Ka*/*Ks*) for each gene, we utilized CODEML in PAML v4.9 [44].

### 2.4. Phylogenetic Analysis

All 56 plastomes and two outgroup species (*Curcuma longa*, NC_042886.1 and *Curcuma phaeocaulis*, NC_045242.1) were included for analysis. Using MAFFT v7.464 [45,46], a whole plastome sequence alignment of 56 plastomes and two outgroup species was produced. The misaligned locations were trimmed using TrimAL v1.3 [47]. According to the annotation files, we extracted and retrieved the longest CDS sequences of 79 genes that code for proteins from the genome sequence of each plastome and compared them using MAFFT [45,46]. To achieve our objectives, we concatenated the alignments of the nucleotide sequence with 79 genes that code for proteins. With the use of 1000 ultrafast bootstrap runs and IQ-TREE v2.0 [44], this data set was also used to reconstruct the phylogenetic tree and evaluate branching support in the FigTree v1.4.3 (http://tree.bio.ed.ac.uk/software/figtree, accessed on 12 March 2023) tree display program.

## 3. Results

### 3.1. General Features of C. alismatifolia Plastomes

Complete plastome lengths for 56 *C. alismatifolia*-cultivated accessions ranged from 162,139 bp to 164,111 bp (Figure 1), Similar to other studies, all the plastomes possessed a typical quadripartite structure including a large single copy (LSC) region (ranging from 86,860–88,364 bp), a small single copy (SSC) region (ranging from 15,692–15,863 bp), and two inverted repeat (IR) regions (ranging from 29,855–30,255 bp). In general, the plastome sequences of the *C. alismatifolia* accessions were similar in length and structure. The average overall GC content among the sequenced plastomes was 36.0% with a 40.9% GC content within IRs, 33.8% within the LSC, and 29.6% within the SSC (Appendix A).

The 56 *C. alismatifolia* accessions were not only similar in length and structure, but also contained the same number of genes. Each accession contained 113 functional genes, with 79 genes that code for proteins, 30 tRNA genes, and four rRNA genes in the plastomes (Appendix A). Among these 113 unique genes, one gene was found that crossed different repeat junctions. The gene *ycf1* extended across the junction of IRA and SSC. Of the 113 genes, the LSC region contained a total of 82 genes including 21 tRNAs and 61 genes that code for proteins. Both inverted repeats contained 20 genes including eight genes that code for proteins, eight tRNAs, and four rRNAs. The SSC region contained13 genes with 11 protein coding genes, one tRNA, and one rRNA. A total of two introns were found in three genes, including *rps12*, *clpP*, and *ycf3*, and one intron in each of the remaining 14 genes, including *trnL-UAA*, *trnA-GUC*, *trnG-UCC*, *trnV-UAC*, *trnI-GAU*, *trnK-UUU*, *rpl2*, *rpoC1*, *atpF*, *rpl16*, *petB*, *ndhA*, *petD*, and *ndhB*. The longest intron of *trnK-UUU* contains the *matK* gene. Trans-splicing of the rps12 gene was predicted to occur with a single 5′ end exon in the LSC and a repetitive 3′ end duplication in two IRs (Appendix A).

### 3.2. Contraction and Expansion of IRs

The total length of the plastome can constrict and expand as a result of differences in the single-copy and IR region sizes at the borders. A total of 11 representative *C. alismatifolia* accessions were compared at the LSC/IR/SSC borders. Despite the four units being very well conserved, the boundary regions of the LSC, IR, and SSC still showed minor differences. For example, the intergenic region linking the IRB and SSC junction displayed a significant range in length among the 11 plastomes tested. Among *trnN-GUU* and *ndhF*, there were approximately 4236 to 4551 bp between them. *Ycf1* spanned the SSC and IRA junction in all 11 plastomes, but there was no gene spanning the IRB and SSC junction. The intergenic regions between the IR and LSC junctions varied significantly among the 11 plastomes with the distance between *rpl22* and *rps19* ranging from 172 bp to 233 bp between the IRB and LSC regions and from 256 bp to 302 bp between *rps19* and *psbA* at the IRA and LSC junction. The main contribution to the difference in IR and LSC boundaries was that the *rps19* gene in accession Ch28 was more distant from *rpl22* than in any other plastome, although the total length of this plastome was shorter than all but 2 of the 11 plastomes compared here (Figure 2).

### 3.3. Sequence Repeats in the Complete Plastome

Nucleotide repeats in plastomes such as SSRs can be particularly helpful markers to recognize species and populations considering the high levels of mutation in these regions. A comparative analysis of repetitive sequences between all 56 plastomes found that the general distribution, amounts of repeats, and types share a lot of similarities among the *C. alismatifolia* and relative species. A total of four SRR types were identified in 56 *C. alismatifolia* plastomes; among these SSRs, 82.7% (3858/4666) were single nucleotide A/T motifs, 15.1% (702/4666) were AT/AT motifs, 0.8% (40/4666) were AAT/ATT motifs, and 1.4% (66/4666) were C/G motifs. Between 55 and 75 A/T SSRs were discovered in the 56 plastomes. Ch63 and Ch47 were found to have fewer than other accessions with only 55 found, indicating that for these accessions, long A/T segments tend to have more indels. The 56 plastomes contained between 10 and 17 AT/AT SSRs, with Ch43 having the highest number. The 56 plastomes were discovered to contain a total of 0–2 C/G SSRs, with Ch52 having none. There were between zero and two AAT/ATT SSRs found in the 56 plastomes. There were 16 accessions found to contain no AAT/ATT SSRs (Figure 3a). The SSR analysis revealed that certain genomic regions can be used to determine the lineages of *C. alismatifolia* if there are data available for various types of SSR motifs and their length variations are combined into a research hierarchy. Through the analysis of SSRs, it has been determined that specific genomic regions can effectively identify the lineages of *C. alismatifolia*. This is achieved by combining data on various types of SSR motifs and their corresponding length variations. In the future, these SSRs might be utilized to provide molecular markers for genetics in population studies and species identification.

All 56 plastomes were analyzed using Reputer to find repeats of a length of at least 8 bp. Four distinct repeats were taken into consideration: forward (F), complement (C), reverse (R), and palindromic (P). The total number for each motif type was determined by grouping them based on sequence length. Most of the repetitive sequences ranged in length from 30 to 39 bp, followed by the 20–29 bp and 40–49 bp ranges, with the >50 bp range having the fewest. In the 20–29 bp group, only three accessions contained all four nucleotide repeats. A total of 43 accessions had three nucleotide repeats not including C repeats, and 10 accessions contained only F and P. In the 30–39 bp group, only 16 accessions contained all four nucleotide repeats. The remaining 40 accessions had three nucleotide repeats not including C. In the 40–49 bp group, nearly half of the 56 accessions contained only F and P. Ch50 and Ch52 contained all four nucleotide repeats, and C repeats were only found in Ch50 and Ch52. In the >50 bp group, 15 accessions contained three nucleotide repeats, and only Ch52 contained four nucleotide repeats (Figure 3b). Based on the variations in the type and quantity of repeats, a creation of markers which might be employed to identify different species or lineages is possible. In previous research, others have found that variations in the location, abundance, and type of repeated sequences found in a genome can serve as reliable indicators to recognize species or populations.

### 3.4. Evolutionary Rates among Protein Coding Genes

To further verify diversity and evolution among functional sequences in *C. alismatifolia* accessions, we estimated the *Ks* for each of 79 genes that code for proteins to compare the rates of evolution between different *C. alismatifolia* plastomes. The most rapidly evolving genes as quantified by *Ks* were *infA* (0.0676), *ndhE* (0.0613), *ndhF* (0.0532), *rpl22* (0.0514), and *petL* (0.0496). By contrast, nearly 20 genes, such as *ndhK*, *psbH*, *rpl32*, *ndhC*, *ycf2*, and *psbF*, the genes which evolved slowly, had *Ks* values close to zero (Figure 4a).

To calculate the evolutionary velocities of various genomic units, these genes were divided into functional or locational (including LSC, SSC, and IR) groupings. When classifying according to their function, transcription genes showed the highest *Ks*; photosynthesis genes showed the lowest *Ka* (Figure 4b). When genes were analyzed based on genomic location, the SSC had the highest rates and the IRs had the lowest (Figure 4c). Most genes had *Ka*/*Ks* values less than 1, indicating that they were susceptible to selective purification (Figure 4b,c). Only two genes had *Ka*/*Ks* values above 1, which were *rps15* (1.15) and *ndhl* (1.13) with *petA* equal to 1 (Appendix A).

### 3.5. Genome Sequence Divergence

A comparison between 11 representative *C. alismatifolia* accessions was conducted to identify mutational hotspots in the plastome. We found that the sequence divergence between different accessions of *C. alismatifolia* was largest in noncoding regions of the plastome. The LSC exhibited a higher level of sequence divergence, and IRs exhibited the lowest. The 11 accessions could be divided into two main types according to their shared plastome variations (Figure 5). However, some hypervariable regions could be detected within each type. For example, the sequences of Ch69, Ch52, Ch61, Ch28, Ch32, and Ch43 are similar in genome sequence divergence, but in the coding regions, fifteen genes possessed greater variability: *rps16*, *psbK*, *atpI*, *rpoC2*, *rpoC1*, *rpl20*, *clpP*, *psbT*, *psbN*, *rpl14*, *ycf2*, *ndhF*, *ndhG*, *ycf2*, *and rps19*. Seventeen intergenic regions also showed higher levels of variations: *trnK-UUU-rps16*, *rps16-trnQ-UUG*, *rps4-trnT-UGU*, *atpI-rps2*, *petN-psbM*, *psbM-trnD-GUC*, *trnY-GUA-trnE-UUC*, *rps4-trnT-UGU*, *ndhC-trnV-UAC*, *rbcL-accD*, *accD-psaL*, *psbE-trnW-CCA*, *psbT-psbN*, *rps12-trnV-GAC*, *ndhF-rpl32*, *ccsA-ndhD*, and *rps15-ycf1*.

### 3.6. Phylogenetic Analyses and Molecular Marker Identification

We completed a phylogenetic analysis utilizing 56 *C. alismatifolia* plastomes to evaluate the divergence of the plastome in the context of evolution and uncover synapomorphies (and eventually barcodes) for certain lineages, with *C. phaeocaulis* (NC_045242.1) and *C. longa* (NC_042886.1) set as outgroups. A phylogenetic tree of *C. alismatifolia* based on all CDSs concatenated the resolved Ch43 with the outgroup *C. phaeocaulis*, suggesting that this accession was either misidentified or of hybrid origin A phylogenetic tree based on concatenated CDSs suggests that the Ch43 accession of *C. alismatifolia* was either misidentified or of hybrid origin, as it resolved with the outgroup *C. phaeocaulis* (Appendix A). The remaining 55 accessions could be further divided into seven clades according to the phylogeny. Ch25 was resolved in a distinct early diverging clade with more support. The color and phenotype of different accessions within clades is diverse, suggesting that morphology and maternal lineage are apparently discordant, but closer examination using the lineages defined here may provide previously undetected morphological similarities (Figure 6).

To further discriminate the 56 *C. alismatifolia* accessions into discrete groupings, we searched for regions that were abundant in SNPs and INDELs to locate possible barcode loci. From the alignment, 25 SNP loci and 6 INDEL loci were discovered in group VI, 145 SNP loci and 69 INDEL loci were discovered in group V, 111 SNP loci and 50 INDEL loci were discovered in group IV, 70 SNP loci and 49 INDEL loci were discovered in group III, 54 SNP loci and 33 INDEL loci were discovered in group II, and 2 SNP loci were discovered in group I (Appendix A). A super DNA barcode advent might be taken into consideration utilizing the complete plastome to recognize *C. alismatifolia* lineages using shotgun sequencing data given the prevalence of SNPs and INDELs throughout the whole genome (Table 1).

## 4. Discussion

In this study, we reported 56 complete plastome sequences from well-known cultivars of *C. alismatifolia*. By assembling genomes and annotating genes, we obtained more particularized information on plastome evolution in *C. alismatifolia* and presented a comparative analysis. From these analyses, it is clear that the overall structure of *C. alismatifolia* plastomes is conserved but that sufficient molecular evolution has occurred such that different groups can be identified.

Contraction and expansion of plastome genomic units and repositioning of IR junctions can result in important evolutionary processes, similar modifications in plastome size, gene duplication, the creation of pseudogenes, or the reduction of many copies of a gene to a single copy [48,49]. Such large boundary differences can be distinct markers for lineage identification [18,24]. We selected 11 representative accessions from different branches of the phylogenetic tree to study IR junctions and sequence evolution. Considerable repositioning in the IR and LSC boundary was found in Ch28 compared to the other accessions (Figure 2). We discovered that there were variations among accessions in IR junction positioning with similar sequences, reflecting the substantial differences within *C. alismatifolia*. Given these length differences at the IR boundaries, molecular markers could be developed at these regions, and through an SNP analysis, we also found some sites that could be used as DNA barcodes [50].

Synonymous mutations are generally considered to not be subjected to natural selection, and subsequently, *Ks* represents the foundational base replacement speed of the evolutionary procedure [51,52,53,54]. We determined the *Ks* for each of the 79 genes that code for proteins in the 56 *C. alismatifolia* accessions to comprehend the evolutionary history of the species (Figure 4). Among them, the *infA* gene had the fastest *Ks* (0.0672). While the majority of the slower-evolving genes were involved in self-replication and photosynthesis [24]. In most cases, the rates of evolution in some plastome genes are species-specific. For example, the *clpP* gene is highly conserved (*Ka*/*Ks* 0.02) in *C. alismatifolia*, but in other angiosperm lineages, it is the most variable gene [16,55]. We discovered that the *clpP* gene contains two introns, which is consistent with previous research and may contribute to the low *Ka*/*Ks* [10,55,56,57,58]. More than half of the *Ka*/*Ks* values of the genes in the *C. alismatifolia* accessions were less than 1, suggesting that they were subject to purifying selection (Figure 4b,c). Only two genes had *Ka*/*Ks* values above 1, which were *rps15* (1.15) and *ndhl* (1.13) with *petA* equal to 1.

In this work, all 56 plastomes were examined to determine the types, quantities, and distribution of repeat sequences. The regions were found to be variable and had the potential to be useful in identifying haplogroups. The phylogenetic tree clades were not clearly associated with flower color, indicating that extensive hybridization of *C. alismatifolia* may have decoupled some morphological characters from maternal inheritance. Interestingly, some phylogenetic patterns derived from functional gene alignments were also reflected in non-functional molecular evolution such as in Ch50 and Ch52, which were resolved on the same branch and possessed a greater number of repeats; they are currently classified as unique *C. alismatifolia* accessions based on these shared, derived sequences (Figure 6). An analysis of the sequence divergence revealed that the intraspecific variation between *C. alismatifolia* accessions was high, suggesting rapid evolution and/or a long time since divergence from sister lineages as well as the possibility that unrecognized cryptic lineages are present. Some morphological characteristics were found to be consistent within a given maternal lineage. Our results demonstrate the potential of plastomic data in resolving maternal relationships in the face of hybridization and polyploidization, which are common in *Curcuma*. The morphological characteristics of different *C. alismatifolia* accessions are very similar to that of other species of *Curcuma*, and from accessions like Ch43, it would appear that interspecific introgression may be contributing to such a morphological homogenization. When nuclear data are available, the data presented in this paper will be of great value in understanding the patterns of maternal introgression in the history of hybrid *Curcuma* cultivation as well as identifying cytonuclear incompatibles.

## Figures and Tables

**Figure 1 genes-14-01743-f001:**
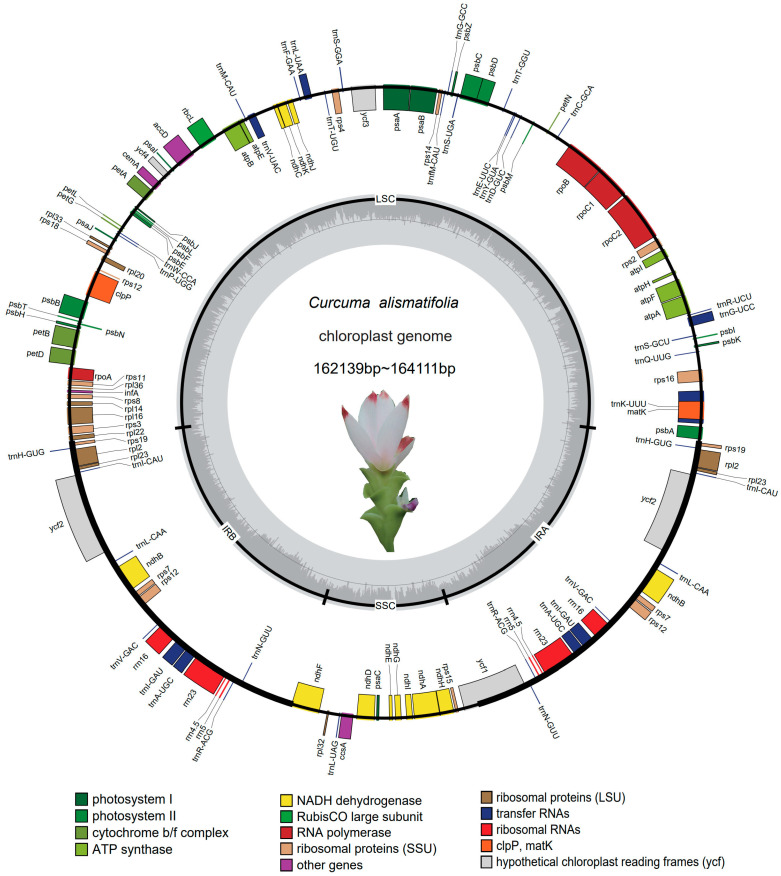
The general plastome structure from 56 *C. alismatifolia* accessions. The inverted repeat regions (IRA and IRB) are identified with thick lines on the outer entire circle. The direction of transcription for the genes outside the circle is clockwise, whereas the genes inside are expressed in a counterclockwise direction. Colors are assigned to genes depending on their functional groupings. Darker/lighter grey bars in the inner ring represent GC/AT content.

**Figure 2 genes-14-01743-f002:**
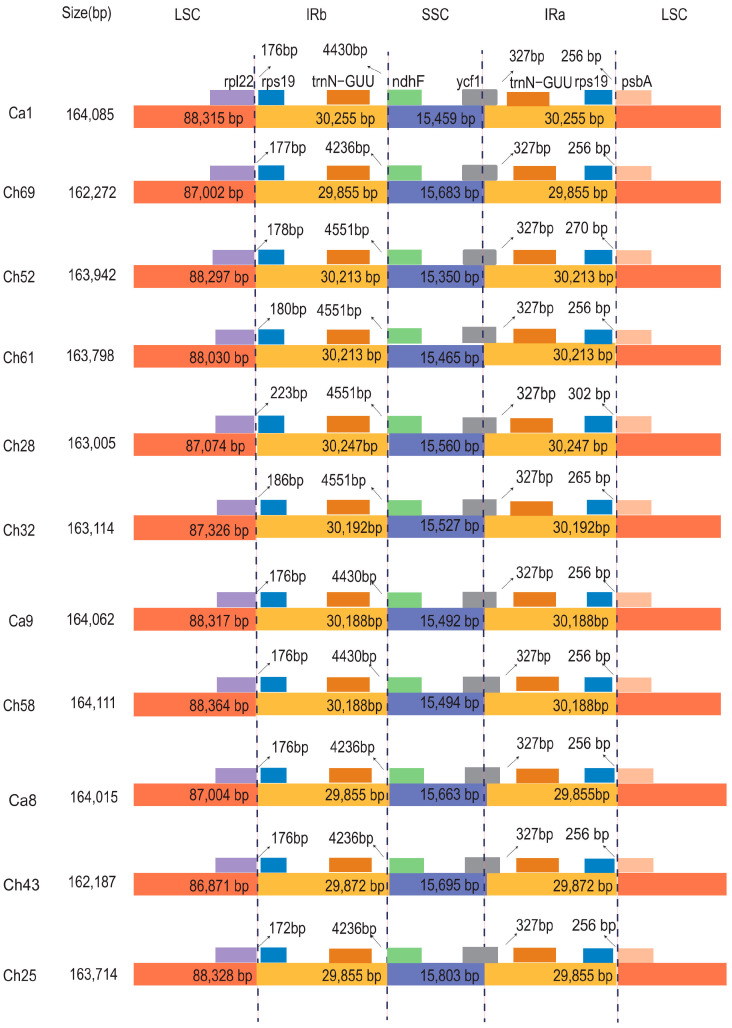
Analyzing 11 *C. alismatifolia* accessions at junctions throughout the LSC, SSC, and IR regions. Figure is not scaled (LSC: large single copy, IRa/b: inverted repeat, SSC: small single copy). The distance between nearby genes is shown via the number displayed next to the straight line at the junction.

**Figure 3 genes-14-01743-f003:**
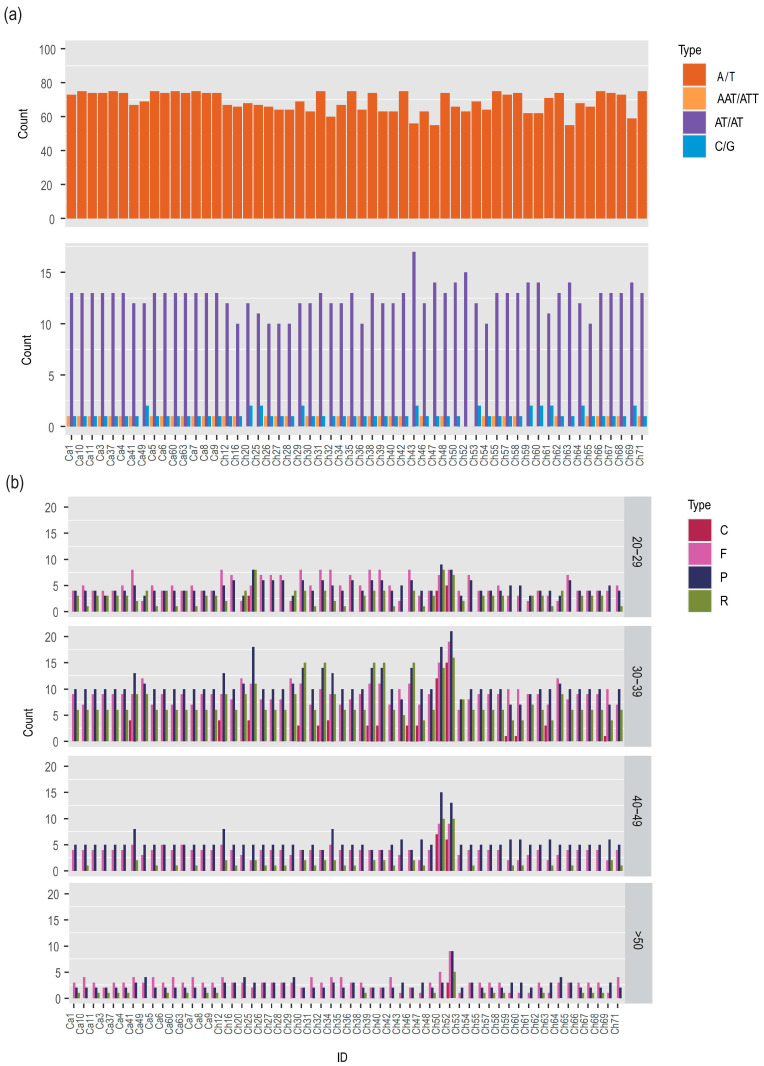
(**a**) The number of distinct types of simple sequences repetition (SSRs) from 56 *C. alismatifolia* plastomes, including A/T, AT/AT, AAT/ATT, and C/G type SSRs. (**b**) Variances in repeat quantity and type in 56 *C. alismatifolia* plastomes including forward (F), complement (C), reverse (R), and palindromic (P) type repeats.

**Figure 4 genes-14-01743-f004:**
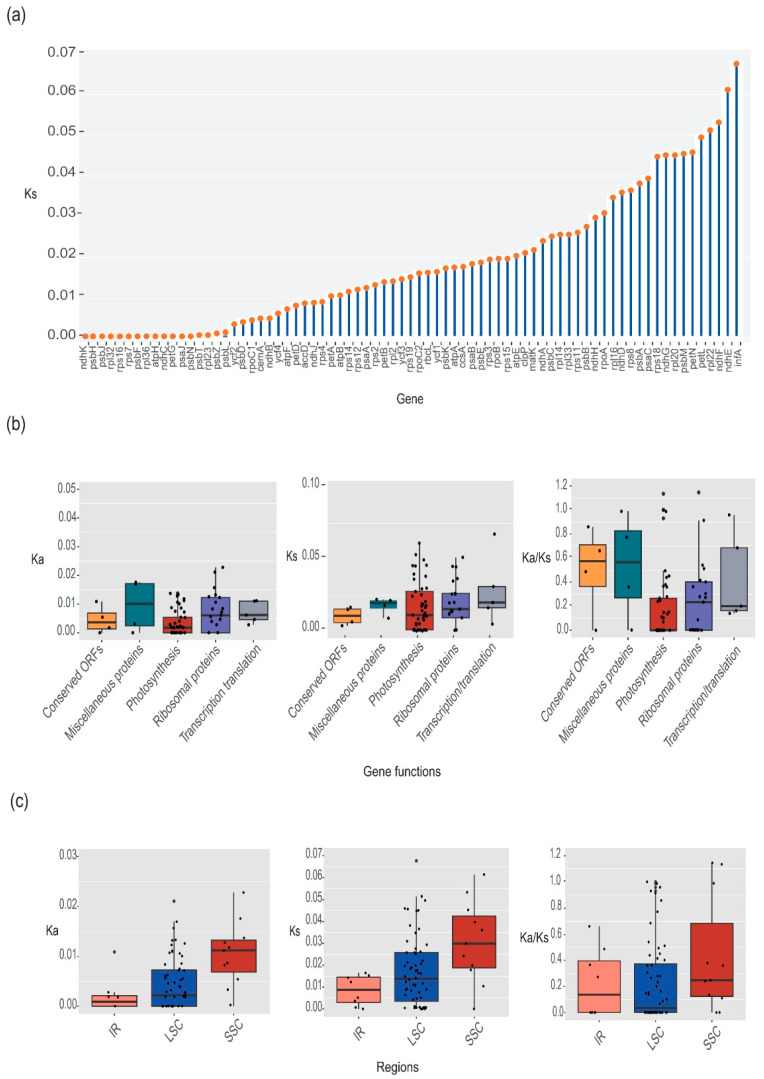
This study examined the selection patterns and intensity of 79 genes that code for proteins in 56 *C. alismatifolia* plastomes. (**a**) *Ks* values of 79 genes that code for proteins ranked by *Ks*. (**b**) *Ka*, *Ks*, and *Ka*/*Ks* ratios in genes with different functional classifications, and (**c**) *Ka*, *Ks*, and *Ka*/*Ks* ratios in LSC, SSC, and IR regions.

**Figure 5 genes-14-01743-f005:**
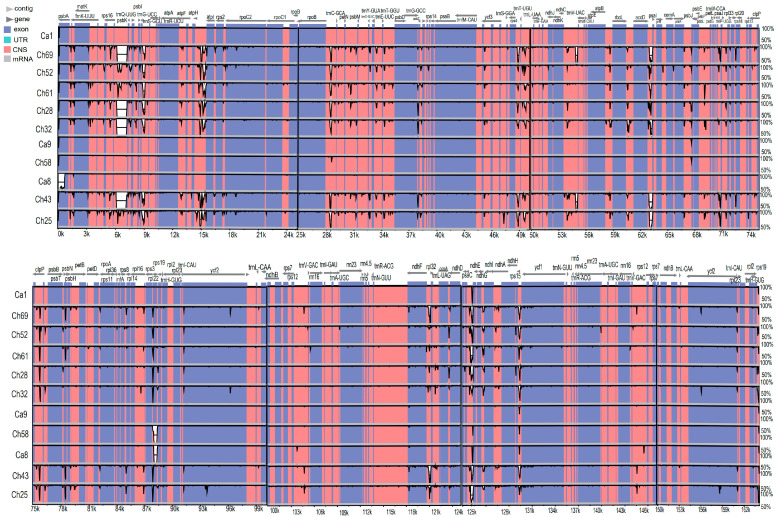
Comparing the sequence divergence of 11 newly assembled *C. alismatifolia* plastomes using mVISTA and Ca1 as reference. The range of sequence identity (50–100%) is shown on the Y-axis. The genes for tRNA and rRNA are not shown in this picture.

**Figure 6 genes-14-01743-f006:**
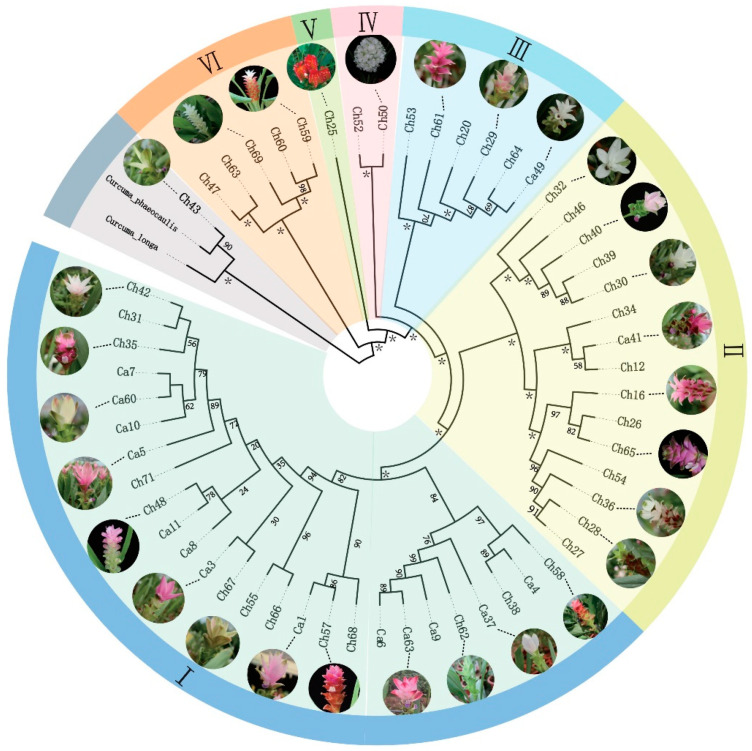
Phylogenetic relationships of 56 *C. alismatifolia* plastomes according to complete plastome sequences. The rapid bootstrap levels produced using IQ-TREE are represented by scores at the nodes; * at nodes represents a bootstrap value of 100%.

**Table 1 genes-14-01743-t001:** Number of potential molecular markers for different branches of the evolutionary tree.

Group	INDELs	SNPs
I	55	100
II	33	54
III	49	70
IV	50	111
V	69	145
VI	6	25

## Data Availability

Not applicable.

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
