# Peer review of "Comparative Plastomes of Curcuma alismatifolia (Zingiberaceae) Reveal Diversified Patterns among 56 Different Cut-Flower Cultivars"

_genes, 2023, doi:10.3390/genes14091743_

Round 1
Reviewer 1 Report
A lot of work has gone into this paper, but while the reporting of plastid data appears to be sound, and indeed excellent, the unavoidable truth is that a plastome phylogeny alone has very little biological meaning for a cultivated taxon where rampant artificial hybridisation has almost certainly occurred. Any systematic investigation of such a species simply must include nuclear data as well. I think perhaps that you needed to think a lot more about the goals of the study, and the questions to be asked, before you began this research. Had you done so, I think you would have seen that a mixture of plastid and nuclear data would have been necessary to understand this particular species.
INTRO: The introduction is poorly focused and very badly structured. Large amounts of information are offered about the anatomy of the study species, which is not directly relevant to the goals of the paper. There is a large paragraph about the utility of the plastome, mainly repeating points I’ve read countless times before. Instead, you need a much briefer piece of text (1-2 sentences) stating why plastome analysis is the appropriate approach for THIS species. Except that, on its own, it is not. The scientific interest in this paper clearly lies with the unravelling of the origin and history of complex cultivars grouped under one species name, and therefore the opening paragraphs would introduce this as a general, conceptual topic. The complete absence of an account of the known history of the target species in cultivation is a major gap. We need to be given a summary of all that is currently known about the time and place of domestication, whether it exists in the wild or was created in cultivation, where the wild relatives occur and/or which species were involved in hybridisation. If the answers to these questions are unknown or uncertain, then you make that clear and use it as a justification for the paper.
METHODS: Other than a failure to give any information on the cultivars sampled, the methods section appears brief but adequate. But this omission is a very big one. Figure six clearly shows the dramatic morphological variation among the sampled individuals, but are we supposed to believe that there is no nominal distinction between any of them? If so it would need to be explained in far more detail because it would be extremely unusual. In Europe or America, every single one of the pictured plants would be going by a different cultivar name. If that’s not happening here, explain why.
RESULTS: The Results section goes into great detail about every aspect of the plastome genomes reported. I am not an expert on this aspect of the paper so will not comment on the quality of the reporting, but my guess is it has been done extremely well. The question that must be asked, however, is what relevance this information has to the questions being asked in the introduction. Sequencing genomes, as an end in itself, is a clear contribution to overall scientific knowledge, but if the goal of this paper is to better understand the study species, then I question whether a paper such as this is the place to report the content of such genomes in detail.
DISCUSSION: The plastome data successfully resolves a series of distinct, apparently monophyletic groups among C. alismatifolia. The Discussion then looks at evolution within the plastome, and the utility of the markers created to distinguish material. Sadly, none of this is really helpful in answering the questions about how this species diverged into so many different forms, in cultivation.
There seem to be two very different papers here that uneasily occupy the same space. The first reports the plastomes recovered confidently and accurately, but without really addressing any scientific questions (e.g. the stuff about genes under selection does not address any of the questions raised in the introduction, although it probably could if the vague point about ‘evolution or genome architecture’ were properly expanded upon), and. Then the second paper attempts to address issues of systematics among the cultivated material, but cannot do this successfully because the plastid alone cannot be enough.
The main issue here is that plastid-based phylogenies, alone, are simply not enough to unravel the history of any lineage where a lot of hybridisation is suspected. Even at the genus level, a plastid tree cannot be considered a species are if members of that genus are known to hybridise a lot. Here, we are dealing with a single species, in a genus which according to the authors “contains many named cultivars with diverse hybrid origins”. The authors successfully identified one accession with the cpDNA of another species - they say C.phaeocaulis and they might be right, but in fact all that can be concluded for certain from their data is that Ch43 is closer to that species than it is to any other accession sampled. Only by sampling the plastomes of all Curcuma species that might have been involved in cultivation creation, can they get a full picture. The might find, for example, that Type VI plastomes also come from a different wild species. This means that all the txt they provide about plastome evolution lacks context - it is not clear how much of the between-individual differences detected are actually between species differences.
A plastome based phylogeny cannot hope to tell us a full story about the evolution in cultivation of C. alismatifolia, because nuclear data will likely paint a very different picture. We cannot even judge which of the many morphological traits that distinguish the sampled accessions (I have only Fig 6 to go on here) are homoplasious, because accessions grouped together for plastids might actually not be similar for nuclear genomes. A proper phylogenetic analysis of nuclear genomes would resolve this far better (there would of course be rampant reticulate evolution, but there are now analysis methods available that are equal to this). The authors could then go further, looking for alleles within the genome that consistently group with certain morphological traits. This could identify genes underlying specific traits, and be useful both for systematic understanding and future cultivar development.
In this context, the current paper feels both incomplete and lacking a clear purpose. My advice to the authors would be to first report the plastomes they have generated in a short paper, leaving out the questions about systematics and the like. The phylogeny they have produced looks good, but has little biological meaning unless nuclear data is brought in as well. If they wanted to add meat to this first paper, they might consider a systematic analysis of morphological traits among the accessions examined, to see if any of these correlate with the plastome phylogeny. If they do, the tentative conclusion would be that either (1) said trait is controlled by a gene in the plastid, or (2) that trait has not been influenced by homoplasy or hybridisation, but both would be tentative. However they approach, the authors must abandon the pretence that phylogeny of plastomes alone has much biological meaning for a cultivated taxon.
I think that an investigation of the phylogenomics of this species would make for a very interesting paper, but I repeat, it simply cannot be done with the plastid alone. Nuclear genome data is essential, and every species that might have been involved needs to be sampled, to build up a clear picture of which genome portions came from where. Likewise, other cultivars of Curcuma not classified as C. alismatifolia should be included as well. Morphological traits should be recorded for every accession and cultivar sampled, in a systematic way, which would allow you to seek homoplasies and identify genes that might underlie such traits. If none of the authors have the systematic expertise for such an analysis, they should involve another who does. I know that all this entails a lot of extra work, but the questions cannot really be addressed without it. I for one would love to read such a paper and think it would fully justify the effort involved.

Generally the quality of English language in this paper is pretty good - well done. One exception is the use of invetered commas to justify a vague word in the second line of the introduction. This may be permissible in popular science writing but it is completely unacceptable in a primary science paper, please never do this again.
The structure of the introduction also needs a lot of work. See uploaded commented MS. Generally a primary paper introduction always starts with broad conceptual background, not the traits of the study organism.
Reviewer 2 Report
The work by Wang et al. harnessed plastomes in 56 accessions of the Curcuma alismatifolia (Zingiberaceae) in order to phylogenetically clarify mixed genetic backgrounds and multiple hybridization events within the species. The authors managed to built comprehensive and robust plastome resources in a hypothesis-driven manner. Overall, the work is well written, statistically up to date, and highlights key findings. However, I have the following major suggestions.
First, the last paragraph of the introduction should explicitly enlist (in L76) the research gap before describing the specific research goals (in L77). Please also close the paragraph with the research hypothesis and the expected results (in L82). This will allow readers focusing on explicit expectations when approaching the in silico analysis. The hypothesis may relate with the intrigue hybridization observed within the species.
Second, in terms of methods, please clarify in L127 that IQ-TREE is a stochastic approach for phylogenetic tree reconstructions using maximum likelihood. Please also comparatively describe the method in comparison with other type of reconstructions, such as Bayesian.
Also methodologically, I recommend authors to implement Approximate Bayesian Computation (ABC) modeling to better reconstruct and date the potential hybridization events. Contrary to phylogenies, ABC embraces and enables considering non-single bifurcation scenarios. Figure 4 in PLoS Genetics 2012 8:e1002703 may inspire authors regarding this novel anlysis.
As closure of the discussion section in L334, please include a perspective section with recommendations on how to better integrate plastome evolutionary analyses to assess across species.
Round 2
Reviewer 1 Report
Most of the changes here feel like sticking plasters, rather than meaningful responses. The fundamental issue, that plastid phylogenies are not really very useful for looking within a cultivated species, remains. I recommend to the authors that they consider restructuring to make this paper about within-species plastid evolution alone (i.e. play to their strengths) and remove any suggestion that it will be of use for understanding relationships within the species (that can be addressed later, with nuclear data). It is very clear that the authors are very strong on the functional genomics of the plastid, but weaker on the issues of cultivar evolution; they also seem to be under pressure from somewhere to publish it very fast. This might be the way to do it.
Introduction: The vague word is gone, thankfully, but the structure issues remain. One piece of text (in yellow) has been added to increase justification for the use of plastids, but the two references offered do nothing to address my comments. They need sources to justify the use of plastids WITHIN a single CULTIVATED species. Even within a single wild species would be a step forward, but even with that they would need to explain why the marker can be useful for a cultivated species. Basically, none of my comments have been dealt with.
My request for info on the sampled cultivars has been completely ignored. The added part about morphology vs clade membership later on makes this even more important (see below)
An acknowledgement has been added at the end that nuclear data is needed, which is welcome. But it is only an acknowledgment of weakness.
The absence of any data on related species remains a weakness. How big a weakness depends on what the eventual focus will be.
I cannot begin to understand why C50 and C52 are classified as “unique C. Alismatifola accessions” (L331-332). The two together form group IV, therefore neither can be described as “unique.” Moreover, every accession sampled is unique in some ways. Basically this sentence does not make sense, and needs to be reworded.
A paragraph has been added describing matches between morphological traits and plastid clade membership, which is very welcome. Unfortunately it is not supported by anything in the Results. I’m guessing the data must exist, based on the existence of this new text, but it needs to be presented. It is not enough to simply report correlations in the discussion without presenting the data. So bract size is uniform in in group IV, but does that make them bigger or smaller that other cultivars? The paper does not say. Can accessions be assigned to this group based on bracts of a certain size range? The paper does not say. Accessions in Group II have “larger bracts” but are these the largEST bracts? The same questions apply. Moreover, there is not enough information here to be actually useful to anybody working with these cultivars. For example, if you want a meaningful output here that is not a description of plastid data, information on how to identify at least some of these clades, based on morphology, would be a good start. The “mottled bracts” is a start, but not clear enough on its own. A TABLE of morphological characters MUST be provided, and from this a systematic account can be provided of what traits match what clade. So there’s been a clear improvement here but it is only a start.
The authors need to think again about the purpose of this paper. A crucial question is, would it be a paper of sufficient interest if it looked only at infraspecific plastid evolution? That would very much play to the paper’s strengths (and clearly, those of the authors, too). It is not for me to decide if this could be of sufficient interest to Genes readers, but the point is, I think that could be done quite easily. However, even so they would need to sample at least some other species (anything that is regularly cultivated, I would suggest) to determine which plastid types are infraspecific and which ones are not. Whole plastids would not be needed, just enough to map the other species onto the phylogeny. Use the plastid data to pick a region that is variable, and sequence the other species for that region. Without such data, even a paper focussed purply on plastid evolution would have a weakness of not knowing how much of the variation on show is within a species.
Alternatively, the authors could try to keep the current focus, but if so they’d need to deal with all of my comments from this review and the previous one.
Reviewer 2 Report
Thanks for the careful improvement, with which I agree
